# Preliminary Single-Cell RNA-Sequencing Analysis Uncovers Adipocyte Heterogeneity in Lipedema

**DOI:** 10.3390/cells13121028

**Published:** 2024-06-13

**Authors:** Andrea Pagani, Dominik Duscher, Sally Kempa, Mojtaba Ghods, Lukas Prantl

**Affiliations:** 1Department of Plastic, Hand and Reconstructive Surgery, University Hospital Regensburg, Franz–Josef–Strauß Allee 11, 93053 Regensburg, Germany; 2Department of Plastic, Aesthetic and Reconstructive Surgery, Clinic Ernst von Bergmann, Charlottenstraße 71, 14467 Potsdam, Germany

**Keywords:** adipose tissue, lipedema, single-cell RNA sequencing, adipocyte

## Abstract

**Background**: Despite its increasing incidence and prevalence throughout Western countries, lipedema continues to be a very enigmatic disease, often misunderstood or misdiagnosed by the medical community and with an intrinsic pathology that is difficult to trace. The nature of lipedemic tissue is one of hypertrophic adipocytes and poor tissue turnover. So far, there are no identified pathways responsible, and little is known about the cell populations of lipedemic fat. **Methods**: Adipose tissue samples were collected from affected areas of both lipedema and healthy participants. For single-cell RNA sequencing analysis, the samples were dissociated into single-cell suspensions using enzymatic digestion and then encapsulated into nanoliter-sized droplets containing barcoded beads. Within each droplet, cellular mRNA was converted into complementary DNA. Complementary DNA molecules were then amplified for downstream analysis. **Results**: The single-cell RNA-sequencing analysis revealed three distinct adipocyte populations at play in lipedema. These populations have unique gene signatures which can be characterized as a lipid generating adipocyte, a disease catalyst adipocyte, and a lipedemic adipocyte. **Conclusions**: The single-cell RNA sequencing of lipedemic tissue samples highlights a triad of distinct adipocyte subpopulations, each characterized by unique gene signatures and functional roles. The interplay between these adipocyte subtypes offers promising insights into the complex pathophysiology of lipedema.

## 1. Introduction

Lipedema is a chronic lipodystrophic disease which predominantly affects women [1,2]. Lipedema could be wrongly classified as a “modern” disease with increasing prevalence in western countries. However, the disproportionate adipose tissue (AT) accumulation in the lower body has been known for a long time, but has not been recognized as a real disease. Dr. Allen and Dr. Hines were the first to describe lipedema more than 70 years ago, and only in 1951 did a seminal paper provide the first insight into this disease [2]. The clinical presentation involves debilitating amounts of swollen, misshapen fat tissue on the limbs and abdomen, which impacts not only the physical, but also the mental well-being of the patients [3,4,5]. Due to lower mobility and greater appearance-related distress, lipedema patients present with a significant reduction in quality of life that often leads to depression [4,6].

The pathophysiology of lipedema is unexplained, meaning that the opportunities for successful treatment are limited. Approximately 11% of women worldwide are left to handle this disabling disease [6]. Despite its prevalence, it is currently unrecognized by insurance companies as a disease. Vyas et Adnan [7] summarized the different stages of lipedema. While stage I presents a normal skin surface with enlarged hypodermis, stage II patients report uneven skin with adipose indentations and large fat masses (lipomas). Stage III is characterized by bulky skin and fat extrusions causing large deformations, especially on the thighs and around the knees. In the final stage (IV), patients develop lipolymphedema, with large overhangs of tissue [Table 1] [8].

The scale accounts for parameters like density and thickening of the subcutis, ‘peau d’orange’, palpable nodules, and any disfigurement arising from advanced lipedema [8]. Lipedema’s morphology refers to its location on the body; the classically affected zones are the lower leg, thigh, buttocks, and arm. The histopathology of lipedema involves dilated lymphatic vessels, hypertrophic adipocytes, increased blood vessel fragility, and notable changes in the metabolite parameters [1,9]. Kempa et al. [10] investigated the use of metabolomics in lipedema research, reporting lower levels of histidine and phenylalanine in lipedemic patients, whereas pyruvic acid was increased compared with the healthy controls. In addition, the authors showed that these differences are likely linked to lipedema rather than body mass index (BMI) variations [11]. While there is not a great deal of fluid gathered in lymphatic tissues, the lymphatic breakdown is a contributing factor to the build–up and accumulation of cells [11,12]. Promising insights were supported by Strohmeier et al. [13], who showed that the soluble factors released from lipedema stromal vascular fraction (SVF) are able to reduce in vitro VE–cadherin levels in endothelial cells. Due to the importance of VE–cadherin in maintaining vascular permeability and in supporting endothelial cell-to-cell junctions, the authors supposed that the downregulation of VE–cadherin expression could be pivotal in lipedema’s pathophysiology. A notable study by Veniaminova et al. [14] utilized single-cell RNA sequencing to delve into the cellular composition of sebaceous glands for the first time, revealing unprecedented details about their cellular diversity and function. This improved method in single-cell analysis has provided deeper and novel insights into the cellular dynamics and molecular pathways involved in acne pathogenesis. Given how the cellular hallmark of lipedema is the hypertrophic adipocyte, it is important to use new techniques without bias to uncover the cell pathophysiology. Single-cell RNA sequencing (scRNA-seq) and downstream cluster analysis have shed light on numerous diseases like Parkinson’s, rheumatoid arthritis, ovarian cancer, and psoriasis.

Lipedemic tissue had not yet been examined using scRNA sequencing [15,16,17,18]. Hence, the authors performed a preliminary scRNA-seq analysis to detect and analyze messenger RNA in lipedemic biological samples with the final goal of identifying novel, lipedema-specific adipocyte subpopulations. This was intended to help and motivate our research group to move forward, as well as to inspire other researchers to use our preliminary data by conducted additional scRNA-sequencing analyses on lipedema adipocytes.

## 2. Materials and Methods

### 2.1. Patient Recruitment and Sample Collection

Human deep subcutaneous AT samples were obtained from 30 donors (15 lipedema patients and 15 healthy patients) undergoing liposuction. Participants had to be 18–45 years of age, obese, affected by lipedema (BMI < 35 kg/m^2^), non–diabetic, and free of any active medical problems. Although, in our inclusion criteria, we had planned to consider both female and male patients, liposuctions were performed only on female patients. The biopsies were performed following institutional ethical guidelines, and participants signed informed consent forms before testing. First, 100–500 mg of the subcutaneous AT of the thigh above the Scarpa’s fascia were collected from both lipedema-affected and healthy individuals. After performing a 7–8 mm incision, a 6 mm Bergström side-cutting biopsy needle (Micrins Surgical, Inc., Lake Forest, IL, USA) was introduced 2.5–3 cm through the skin incision into the deeper adipose layers. Brief suction from a 60 cc syringe attached to the needle with gastrointestinal irrigation tubing was applied (Kendall; no-1; 16 Fr/CH × 48 inches, Mansfield, MA, USA). Ultrasound guidance (Toshiba Aplio 500; Toshiba Corporation, Tokyo, Japan) and color Doppler imaging were used to ensure the safety of the procedure. To reduce the discomfort of the procedure, the Scarpa’s fascia was anesthetized with 1% lidocaine.

### 2.2. Adipose Tissue Homogenization, Enzymatic Cell Dissociation, and Single-Cell RNA Sequencing

After sample harvesting, AT samples were kept on ice until processing (Gibco Recovery Cell Culture Freezing Medium, 10%DMSO). After washing the AT 3 times with 1 × PBS, we performed a physical homogenization in adipose harvest media (1 × HBSS + MgCl, CaCl + 5% heat–inactivated FBS, 1% L–glutamine and penicillin–streptomycin, 50 microg/mL DNAse 1) using a blender (Ninja BL450 Series). After homogenization, the sample was distributed into 14 mL tubes and supplemented with Collagenase Type II digestion media (Worthington Biochem, Lakewood, NJ, USA) before shaking and incubation at 37° for 1 h. Digested samples were then filtered and centrifuged to isolate the SVF. To ensure the inclusion of the adipocytes which were buoyant and traditionally separate during centrifugation in our single-cell analysis, we integrated both the buoyant and pellet fractions by collecting the floating adipocytes immediately after the initial centrifugation, following by the above-mentioned secondary step, where the buoyant fraction was digested to further dissociate any remaining cell clusters. The sample was then split, resuspended in a mixture of 9:1 FBS to DMSO solution, and frozen at −80 °C. For scRNA-seq analysis, individual cells were encapsulated into nanoliter-sized droplets (Drop-seq method) containing barcoded beads using a microfluidic device. Each bead carried a single DNA barcode to uniquely identify the source cell.

### 2.3. Complementary DNA Synthesis and Amplification

Within each droplet, cellular mRNA was captured by the barcoded beads and converted into complementary DNA (cDNA). According to the manufacturer’s protocol, cell capture, cDNA amplification and scRNA–seq libraries were generated with the Chromium Single cell 3’ Library and Gel Bead Kit (v2 10× Genomics). cDNA molecules were subsequently amplified via polymerase chain reaction (PCR) to generate sufficient material for downstream sequencing analysis. Libraries were sequenced using the NovaSeq 6000 S2 platform (Illumina, San Diego, CA, USA). Raw reads were aligned to the human genome (hg38) and cells were called using cellranger count (v3.0.2).

### 2.4. Data Processing and Statistical Analysis

Raw reads were aligned to the human genome (hg38), and cells were called using cellranger count (v3.0.2). Cell barcodes with fewer than 200 and more than 13,750 unique molecular identifiers (UMIs) per cell were extracted from the reads, allowing for the assignment of sequencing reads to individual cells and quantification of gene expression levels. Bioinformatics analysis pipelines, including dimensionality reduction techniques and differential gene expression analysis, were applied to compare the transcriptomic profiles of cells from lipedema patients and healthy samples. Statistical tests, such as *t*-tests and Wilcoxon rank-sum tests, were performed to identify differentially expressed genes and characterize transcriptional differences between lipedema and healthy samples. Multiple testing corrections were applied to control for false discovery rates.

## 3. Results and Discussion

As Buso et al. [19] outlined recently, lipedema researchers and plastic surgeons are being called to action to define lipedema’s epidemiology, pathophysiology, differential diagnosis, and management. In this paper, we determine how single-cell RNA analysis could shed light on these challenging and open questions.

Adipocyte heterogeneity has thus far consisted of brown, white, and beige adipocytes [20]. However, heterogeneity across white adipocytes is an emerging field, especially when considering lipedemic adipose tissue. One study showed, for example, a significant difference in the nutrient uptake of healthy white fat in Rhesus macaques which was not dependent on cell size, implying intrinsic heterogeneity. This cell heterogeneity is lost in obese white fat [21]. More recently, in vitro and in vivo clonal analyses of white fat in healthy mice revealed three distinct adipocyte populations, with unique gene expression profiles chiefly surrounding the expression of Wt1, Tgln, and Mx1 [22]. Our data from human lipedema tissue show a triad of different adipocytes with their own gene signatures and functional roles. Isolation and interrogation of particular genes differentially expressed in the clusters of adipocytes can better illustrate lipedema’s pathology. We identified two distinct adipocyte populations (Adipocyte A and B) in both lipedemic and non-lipedemic fat, as well as an enrichment of myeloid cells associated with severe lipedema (Adipocyte C) [Figure 1].

*Adipocyte A* cells were found chiefly in the lipedemic fat (85%), and only in very small proportions in the non-lipedemic fat (15%). Their specific gene profile is mainly represented by Apolipoprotein D (ApoD), Matrix gla protein (MGP), insulin–like growth factor (IGF), aldehyde dehydrogenase 1 (ALDH1A1), and chemochine (C-X-C 14) ligand 14. ApoD is a 29 kDa glycoprotein that is part of human plasma. It is mainly known to stimulate lipid transportation and is itself a component of HDL [18]. Because of this, ApoD is highly upregulated in nerve crush injuries in order to transport myelin after injury. MGP and IGF are stimulators of adipogenesis [23,24]. Meanwhile, ALDH1A1 has a clearly defined role in lipid metabolism [25], and CXCL14 is active in glucose metabolism and macrophage recruitment into the fat tissue [26]. Altogether, this panel of genes suggests that the Adipocyte A population is actively stimulating adipogenesis, the transport of lipid, and lipid metabolism.

In comparison with Adipocyte A, Adipocyte B cells have a different gene signature, and interestingly, were found to be spread across the lipedemic and control fat in a 50:50 ratio (Figure 1). CD55 is highly expressed by Adipocyte B; it accelerates the decay of C3 convertase, thereby limiting the formation of membrane attack complex (MAC) [27]. A blockade in the complement pathway will lead to a build-up of fluid and defective cells in the tissue [28]. Adipocyte B also expresses Semaphorin 3C (SEMA3C), an adipokine secreted by adipocytes of the fat tissue of overweight males [29]. In women, SEMA3C induces the collapse of the lymphatic endothelial cytoskeleton and prohibits the proliferation of lymphatic endothelial cells [30]. The lymphatic vessels in lipedema are described as over-dilated [9], which would be explained by a cytoskeletal breakdown stimulated by SEMA3C. The microaneurysms of the lymphatic system in lipedema, leading to uncontrolled fluid levels in the tissue, could be fueled by Adipocyte B signaling [31]. Finally, even Versican (VCAN) and the secreted frizzled protein 4 (SFRP4) are expressed highly by Adipocyte B. While VCAN is known to sequester and bind high-density lipoproteins from serum and store them in the tissue, SFRP4 is a soluble WNT pathway modulator that plays a role in adipogenesis and adipocyte differentiation [32]. As Han et al. [33] recently explored, the WNT/β-catenin signaling pathway has also been highlighted as a crucial regulator in the accumulation of hair follicle stem cells and the filling of sebaceous glands. The group demonstrated that activation of this pathway facilitates the migration of hair follicle stem cells into sebaceous glands, where they can differentiate into sebaceous cells. This process not only plays a role in normal gland function, but also has therapeutic implications for wound healing. Considering the roles of these genes and contextualizing them together, Adipocyte B has the power to facilitate lipid, defective cell, and fluid storage by destabilizing the microlymphatic system and hindering phagocytosis. In normal, healthy fat, there is likely a homeostatic role for this population, which is the reason for the equal amount of Adipocyte B in healthy and lipedemic fat. However, in lipedemic fat, it is possible that Adipocyte B acts as a catalyst for disease progression. Altogether, despite these promising evaluations and the equal proportion in both controls and lipedema patients, we cannot consider Adipocyte B as primarily responsible for the lipedema. At present, we do not have enough data or results to make a solid evaluation of its role in lipedema. The nature of the adipocyte has some characteristics that could catalyze or induce the disease, but the data we have collected are not sufficient to make solid or certain considerations.

*Adipocyte C* was exclusively found in lipedemic fat, and not in the healthy controls. The topmost highly expressed genes, fatty acid binding protein 4 (FABP4), CD36, retinoid binding protein (RBP7), serum deprivation-response protein (SDPR), alpha 2 macroglobulin (A2M), and interferon alpha-inducible protein 27 (IFNI27), appear to be made of large [34], hypertrophic cells that are still maintaining a level of metabolic activity [35,36,37]. Since Adipocytes C were exclusively found in the lipedema samples, it is likely they make up the bulk of the disease tissue.

The single-cell RNA-seq allowed us to additionally produce a Kyoto Encyclopedia of Genes and Genomes (*KEGG*) metabolic pathway of both adipocytes A and B as a computer representation of their biological system [Figure 2].

## 4. Summary and Conclusions

By taking a deeper look at the proportion analysis, Adipocytes A and C appear to be disease-specific. Adipocyte C is the most abundant of all adipocytes, and its genetic signature gives rise to adipocytes holding high quantities of lipids while staying simultaneously viable: the lipedemic adipocytes [Figure 1]. Based on this evidence, the chain of cell reactions could resemble lymphatic vessels weakened by Adipocyte B being taxed with abundant hypertrophy and hyperplasia of Adipocyte C, facilitated by Adipocyte A. This microenvironment of AT dystrophy contains three distinct clusters of adipocytes which, together, explain the cellular pathophysiology of lipedema (Figure 3).

The adipocyte cluster A is the “lipid generation population”, expressing genes that produce a significant amount of lipids and taking care of their transport. They provide lipids to cluster C and stimulate proliferation. Adipocyte B is the “disease-enabling population” and expresses genes that impact the structure of the tissue by simultaneously preventing phagocytosis of dead cells. This causes a collapse of the micro-lymphatic system of the area and binds high-density lipoproteins to the microenvironment. The final group is represented by Adipocyte Cluster C, the “disease cell population”, which is influenced by the subgroups A and B by holding large amounts of lipid and simultaneously remaining viable (Figure 3).

The application of single-cell genomics techniques has highlighted the identification of previously unrecognized heterogeneity within adipocyte populations, particularly in the context of lipedema. This study unveils a triad of distinct adipocyte subpopulations, each characterized by unique gene signatures and functional roles. While Adipocyte A, predominantly present in lipedemic fat, exhibits a gene profile suggestive of active involvement in adipogenesis, lipid transportation, and metabolism, Adipocyte B displays gene expression patterns associated with complement pathway regulation, lymphatic dysfunction, and lipid storage, potentially indicating a role in disease progression within lipedemic tissue. Adipocyte C, nearly exclusive to lipedemic fat, demonstrates a genetic signature consistent with large, hypertrophic cells with sustained metabolic activity, likely representing the hallmark lipedemic adipocyte. The interplay between these adipocyte subtypes offers valuable insights into the complex pathophysiology of lipedema. We hope that these cellular findings will push lipedema research further, helping other authors to elucidate the underlying mechanisms contributing to the development and progression of lipedema. Further exploration of the genes and pathways identified within these distinct adipocyte clusters holds promise for uncovering novel therapeutic targets and refining treatment approaches for individuals affected by lipedema.

## Figures and Tables

**Figure 1 cells-13-01028-f001:**
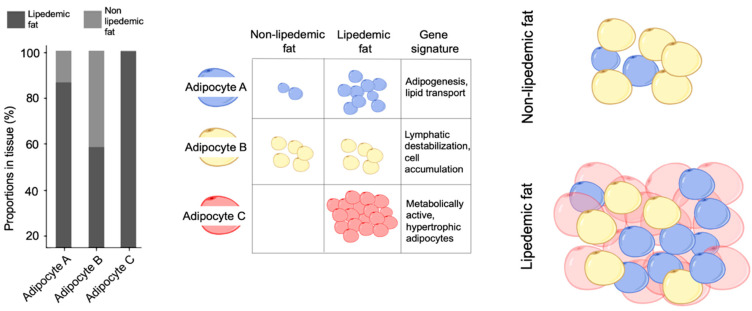
Adipocyte subpopulations, distribution, and gene signature. Whereas Adipocyte A supports adipogenesis and lipid transport and are mainly distributed in the lipedemic fat, Adipocyte B is responsible for lymphatic destabilization and is spread across both lipedemic and control fat. Adipocyte C has an active metabolism and is mainly found in the hypertrophic component of the lipedemic fat.

**Figure 2 cells-13-01028-f002:**
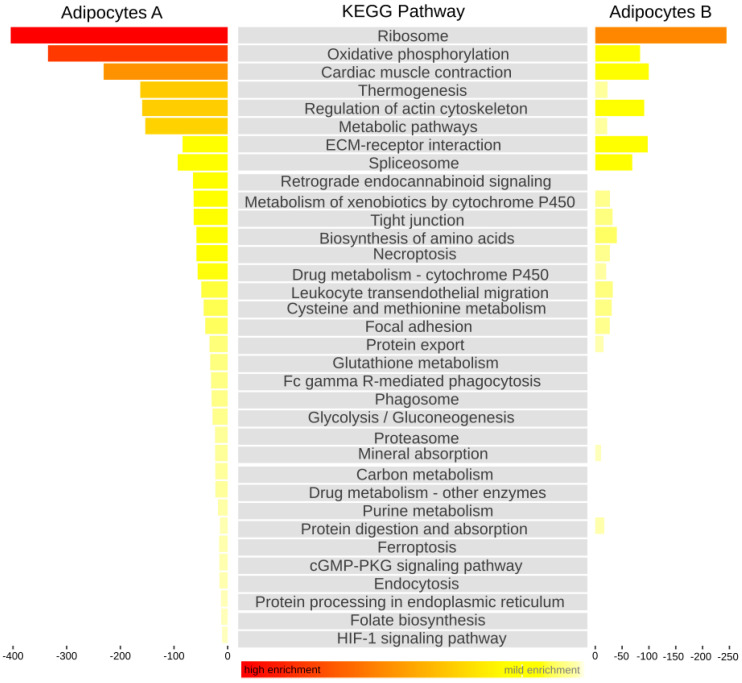
The Kyoto Encyclopedia of Genes and Genomes (KEGG) Pathway. The metabolic pathways of both adipocyte A and B were compared in order to analyze gene functions and link genomic information with higher-order functional information.

**Figure 3 cells-13-01028-f003:**
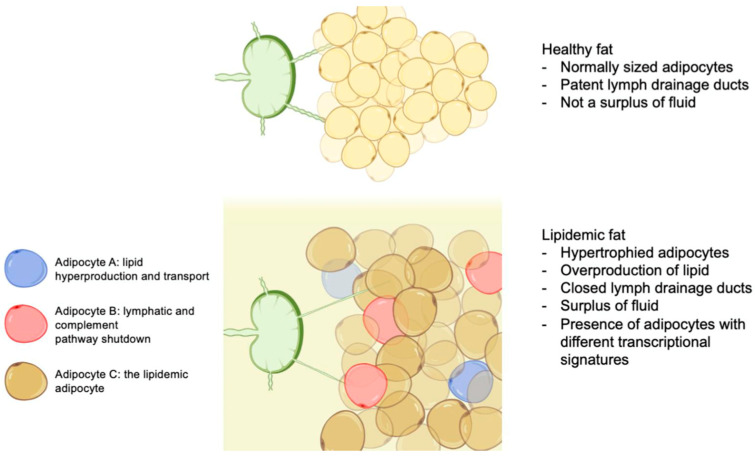
A cellular ethiopathological theory of lipedema formation. Lymphatic vessels weakened by Adipocyte B are taxed with abundant hypertrophy and hyperplasia of Adipocyte C, facilitated by Adipocyte A. This microenvironment of AT dystrophy contains three distinct clusters of adipocytes which, together, explain the cellular pathophysiology of lipedema.

**Table 1 cells-13-01028-t001:** Types and stages of lipedema. Depending on the extremity (lower or upper part) there are different types (I–IV) and stages (I–IV) of lipedema.

LIPEDEMA CLASSIFICATION: STAGES (I–III) AND MORPHOLOGY (I–V)					
**STAGES**	**Stage I** **Thickened but soft subcutaneous tissue with small palpable nodules; smooth skin surface**	**Stage II** **Thickened subcutaneous tissue, soft; some larger nodules; uneven skin surface.**	**Stage III** **Thickened subcutaneous tissue; indurated, large nodules and formation of folds**		
**MORPHOLOGY**	**Type I:** Buttock	**Type II:** Thigh	**Type III:** Entire leg	**Type IV:** Arm (often associated withType II/III)	**Type V:** Lower Leg

## Data Availability

The data presented in this study are available on request from the corresponding author.

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
