# Peer review of "Preliminary Single-Cell RNA-Sequencing Analysis Uncovers Adipocyte Heterogeneity in Lipedema"

_cells, 2024, doi:10.3390/cells13121028_

Round 1

Reviewer 1 Report

Comments and Suggestions for Authors

In this manuscript, the authors conducted an investigation into the cellular characteristics of lipedema utilizing scRNA-seq analysis. Given the absence of a well-established pathophysiology for lipedema, the identification of cellular characteristics of adipocytes, which constitute the major components of lipedema, holds significant importance. In this regard, the utilization of novel techniques such as scRNA-seq analysis is deemed appropriate for discerning adipocyte heterogeneity in lipedema. Additionally, the authors discovered three distinct adipocyte populations in lipedema and elucidated the differential gene expression and metabolic reprogramming, representing the novelty of this manuscript.

Despite the novelty of the findings and the inclusion of preliminary analysis, there is a need to address the detailed methods employed in this study to facilitate the interpretation of the authors’ findings and conclusions. For instance, clarification is required regarding the rationale behind selecting scRNA-seq analysis over snRNA-seq analysis (single nucleus RNA-seq analysis) for exploring adipocyte heterogeneity, considering that scRNA-seq analysis using Chromium may not generate transcriptomics data directly from the larger and more fragile adipocytes themselves. Another aspect to consider is the methodology for identifying adipocyte populations among Stromal Vascular Fractions (SVFs) from lipedema.

Furthermore, it is essential to elaborate on the interplay among different cell populations in lipedema, particularly focusing on the interactions among the three identified adipocyte populations. Addressing these considerations would not only enhance the manuscript but also contribute to expanding our understanding within the research community.

Author Response

  1. Despite the novelty of the findings and the inclusion of preliminary analysis, there is a need to address the detailed methods employed in this study to facilitate the interpretation of the authors’ findings and conclusions. For instance, clarification is required regarding the rationale behind selecting scRNA-seq analysis over snRNA-seq analysis (single nucleus RNA-seq analysis) for exploring adipocyte heterogeneity, considering that scRNA-seq analysis using Chromium may not generate transcriptomics data directly from the larger and more fragile adipocytes themselves. […].

As reported by several research papers but, above all, from Ding et al. [1], the choice between scRNA–seq and snRNA–seq for exploring adipocyte heterogeneity depends on several factors related to cell size, fragility, and transcriptomic fidelity. ScRNA-seq is particular important to enable the profiling of individual adipocytes which have a high cytoplasmic volume and diverse transcriptome due to their metabolic activities because it provides a comprehensive representation of the cytoplasmic RNA, including mRNA and non-coding RNAs. Platforms like the 10x Genomics Chromium system improved in capturing and sequencing larger cells, enhancing the viability and recovery of fragile cells, including adipocytes from lipedema patients, despite their size and delicate structure. As the reviewer mentioned, lipedema–adipocytes are large and fragile, which makes them difficult to capture using standard scRNA-seq protocols. However, our dissociation technique and accurate handling methods in the laboratory reduce cell lysis and improve the recovery of intact adipocytes. Despite the bias in scRNA-seq towards smaller cells, we focused on larger adipocytes, ensuring that their unique transcriptomic profiles are not underrepresented.

On the other side, snRNA-seq is also advantageous for fragile cells like adipocytes or for tissues that are difficult to dissociate into single cells, as it isolates nuclei rather than whole cells, reducing cell lysis and providing a more stable RNA source. However, to the present date snRNA-seq was mainly used for retrospective studies and comparisons across different conditions or time point, in case of nucleus isolation with frozen or archived samples, which can be beneficial when fresh tissue is not available, and this is not our case. On the other hand, we agree with the author that snRNA-seq reduces technical biases related to cell size and structure, that helps to obtain a more uniform dataset, particularly when dealing with heterogeneous tissues like the lipedemic but even the healthy adipose tissue [2]. Recently, Kim et al. [3] showed that while snRNA-seq provides robust data from fragile and large cells, it often underrepresents certain cytoplasmic transcripts and RNA species abundant in the cytoplasm. This can limit the ability to fully capture the functional state of adipocytes, which rely heavily on cytoplasmic processes for their metabolic functions.

Altogether, considering the current literature [4] but even our laboratory experience with scRNA–seq, the decision to use scRNA-seq over snRNA-seq for exploring adipocyte heterogeneity hinges on the need for comprehensive transcriptomic data that includes cytoplasmic RNA species. Although snRNA-seq offers advantages in handling fragile cells [2], scRNA-seq provides a richer and more detailed view of adipocyte heterogeneity, especially for lipedema samples.

REFERENCES: 

  1. Ding J, Adiconis X, Simmons SK, et al. Systematic comparison of single-cell and single-nucleus RNA-sequencing methods. Nat Biotechnol. 2020;38(6):737-746. doi:10.1038/s41587-020-0465-8
  2. Haque A, Engel J, Teichmann SA, Lönnberg T. A practical guide to single-cell RNA-sequencing for biomedical research and clinical applications. Genome Med. 2017;9(1):75. doi:10.1186/s13073-017-0467-4
  3. Kim N, Kang H, Jo A, Yoo SA, Lee HO. Perspectives on single-nucleus RNA sequencing in different cell types and tissues. J Pathol Transl Med. 2023;57(1):52-59. doi:10.4132/jptm.2022.12.19
  4. Vijay J, Gauthier MF, Biswell RL, et al. Single-cell analysis of human adipose tissue identifies depot and disease specific cell types. Nat Metab. 2020;2(1):97-109. doi:10.1038/s42255-019-0152-6

  1. Another aspect to consider is the methodology for identifying adipocyte populations among Stromal Vascular Fractions (SVFs) from lipedema. […]

The reviewer asks the authors for a clarification about the following section of the manuscript:
[…] Digested samples were then filtered and centrifuged to isolate the SVF. The SVF was split and resuspended in a mixture of 9:1 FBS to DMSO solution and frozen at -80°C. For scRNA–seq analysis, individual cells were encapsulated into nanoliter-sized droplets (Drop–seq method) containing barcoded beads using a microfluidic device. Each bead carried a single DNA barcode to uniquely identify the source cell. […]“

Cryopreservation represents a well-established for maintaining cell viability and integrity during freezing and thawing processes [5]. This is particularly important in our preliminary analysis for adipocytes, which are large and fragile. This method ensures that adipocytes retain their functional and structural properties, making them suitable for subsequent scRNA-seq analysis. Storing cells at -80° allowed us to collect samples at different time points and batch process them later. This reduces the variability that can arise from processing samples at different times and improves experimental consistency and reproducibility. In addition, Drop-seq technology enables the high-throughput analysis of thousands to tens of thousands of individual cells simultaneously. This capacity is essential for capturing the full diversity of adipocyte populations and identifying rare cell types that may be missed with lower-throughput methods [6.] While, barcoded beads ensure that each adipocyte's RNA is uniquely tagged, preventing cross-contamination and allowing for precise tracking of the transcriptome from individual cells (critical for accurately identifying and characterizing different adipocyte populations), the encapsulation in nanoliter-sized droplets using a microfluidic device minimizes technical variability and potential biases. The controlled and reproducible nature of microfluidic encapsulation ensures consistent cell handling and processing, which is key for reliable scRNA-seq data. Especially in our adipocyte analysis, the Drop-seq technology is highly scalable and adaptable, making it suitable for our study. It is (relatively) cost-effective compared to other single-cell sequencing methods, especially when processing large numbers of cells. Considering all these aspects, our proposed method provides a robust, flexible, and comprehensive approach to studying adipocyte heterogeneity in Lipedema.

REFERENCES:

  1. Whaley D, Damyar K, Witek RP, Mendoza A, Alexander M, Lakey JR. Cryopreservation: An Overview of Principles and Cell-Specific Considerations. Cell Transplant. 2021;30:963689721999617. doi:10.1177/0963689721999617
  2. Ziegenhain C, Vieth B, Parekh S, et al. Comparative Analysis of Single-Cell RNA Sequencing Methods. Mol Cell. 2017;65(4):631-643.e4. doi:10.1016/j.molcel.2017.01.023
  3. Furthermore, it is essential to elaborate on the interplay among different cell populations in lipedema, particularly focusing on the interactions among the three identified adipocyte populations.

    Thank you for putting particular attention to this part of the manuscript. As widely analyzes in the section “Results and Discussion” our study highlights the complex interplay among three distinct adipocyte populations in lipedema. While Adipocytes A should primarily promote adipogenesis and lipid metabolism, contributing significantly to fat accumulation in lipedema, Adipocytes B could be the main protagonists that disrupt lymphatic function, leading to fluid retention and tissue edema, while also facilitating lipid storage. Adipocytes C are enriched in lipedemic tissue, maintaining metabolic activity and contributing to inflammation. These interactions underscore the multifaceted nature of lipedema pathology, with each adipocyte population playing a unique role in disease progression. This understanding is crucial for developing targeted therapeutic strategies and begin new deeper studies on adipocytes heterogeneity in Lipedema. Considering that our study is a preliminary analysis of adipocyte populations and the considerations are already made on a small number of patients, we believe it is excessive to proceed with further considerations on possible interactions between the populations.

Reviewer 2 Report

Comments and Suggestions for Authors

The study explores the heterogeneity of adipocytes in lipedema through single-cell RNA sequencing analysis. It identifies three distinct groups of adipocytes within lipedemic tissue, each characterized by unique gene traits that play different roles in the pathophysiology of the disease. This research is crucial as it reveals complex cellular interactions in lipedema, setting the stage for targeted treatments. However, the article has some issues:

  1. There are significant formatting problems with Table 1.
  2. The theme of the review is very interesting, but it could be improved by mentioning recent breakthroughs in the field of dermatology related to single-cell sequencing. Including such information would enhance the analysis.

The article mainly discusses the heterogeneity of adipocytes in lipedema, discovering three different types of adipocytes through single-cell RNA sequencing technology:Adipocyte A is predominantly found in lipedemic fat, and its gene profile supports fat production, lipid transport, and metabolism.Adipocyte B is present in both lipedemic and non-lipedemic fat, expressing genes related to the regulation of the complement pathway, lymphatic dysfunction, and lipid storage.Adipocyte C is almost exclusively found in lipedemic fat, showing gene expression related to large, hypertrophic cells with ongoing metabolic activity. Each type has unique gene expression traits and functional roles. The article emphasizes the role of these adipocyte subtypes in the pathophysiology of lipedema, providing new insights into the complex pathological mechanisms of lipedema. This part of the article resembles recent publications, suggesting that additional insights should be considered. There have been modest breakthroughs in the fundamental research of acne as of 2023, including single-cell data on sebaceous glands that have improved the method in single-cell analysis (Veniaminova, N., Cell Rep 2023). Furthermore, the WNT/β-catenin pathway plays a significant role in facilitating the accumulation of hair follicle stem cells and the filling of sebaceous glands (Han, J., iScience 2023). When populated within these glands, these cells have the potential to evolve into sebaceous cells, exerting a beneficial influence on wound healing (Han, J., iScience 2023; Veniaminova, N., Cell Rep 2023). These findings help further understand the cellular basis of lipedema and potential therapeutic targets.

Comments on the Quality of English Language

Minor editing of English language required.

Author Response

The study explores the heterogeneity of adipocytes in lipedema through single-cell RNA sequencing analysis. It identifies three distinct groups of adipocytes within lipedemic tissue, each characterized by unique gene traits that play different roles in the pathophysiology of the disease. This research is crucial as it reveals complex cellular interactions in lipedema, setting the stage for targeted treatments. However, the article has some issues:

  1. There are significant formatting problems with Table 1.

    Thank you for considering our paper for publication into Cells. We agree and we therefore provided a new table in order to have the right format inside of our paper.

  2. The theme of the review is very interesting, but it could be improved by mentioning recent breakthroughs in the field of dermatology related to single-cell sequencing. Including such information would enhance the analysis. There have been modest breakthroughs in the fundamental research of acne as of 2023, including single-cell data on sebaceous glands that have improved the method in single-cell analysis (Veniaminova, N., Cell Rep 2023). Furthermore, the WNT/β-catenin pathway plays a significant role in facilitating the accumulation of hair follicle stem cells and the filling of sebaceous glands (Han, J., iScience 2023). When populated within these glands, these cells have the potential to evolve into sebaceous cells, exerting a beneficial influence on wound healing (Han, J., iScience 2023; Veniaminova, N., Cell Rep 2023). These findings help further understand the cellular basis of lipedema and potential therapeutic targets.

As mentioned from the reviewer, recent breakthroughs in acne research as of 2023 have significantly advanced our understanding of sebaceous gland biology and potential therapeutic targets. A notable study by Veniaminova et al. [7] utilized single-cell RNA sequencing to delve into the cellular composition of sebaceous glands, revealing unprecedented details about their cellular diversity and function. Furthermore, the WNT/β-catenin signaling pathway has been also highlighted as a crucial regulator in the accumulation of hair follicle stem cells and the filling of sebaceous glands [8]. These findings provide a valuable framework for understanding the cellular mechanisms underlying acne and related disorders. They also underscore the potential for targeting specific cellular pathways, such as WNT/β-catenin, in developing innovative treatments. We cited this two studies along the manuscript [7; 8].

REFERENCES:

  1. Veniaminova NA, Jia YY, Hartigan AM, et al. Distinct mechanisms for sebaceous gland self-renewal and regeneration provide durability in response to injury. Cell Rep. 2023;42(9):113121. doi:10.1016/j.celrep.2023.113121
  2. Han J, Chu W, Lin K, Wang X, Gao Y. β-catenin activation in Gli-1+ stem cells leads to reprograming of the hair follicle. Eur J Dermatol. 2023;33(6):710-712. doi:10.1684/ejd.2023.4621

Reviewer 3 Report

Comments and Suggestions for Authors

COMMENTS AND SUGGESTIONS FOR AUTHORS

1.    Please re-size table 1.

2.    In line 152, please change “insulin growth factor” to “Insulin-like growth factor”.

3.    Please include t-SNE or UMAP plots showing subpopulations in control and lipedemic patients in figure 1.

4.    In figure 1, the graph appears misleading in the sense that it is conveying information that adipocyte C makes up 100% of the population in lipedema tissue, even though that is not the case. Please re-phrase the related description and re-make the graph.

5.    Was there a difference between males and females regarding these populations? Were both sexes included in the study?

6.    In summary and conclusion, the claim that adipocyte B is disease enabling population is not backed by results, since there is equal proportion of this population in both controls and patient subjects. It is difficult to be convinced that the population can behave in different ways in different conditions in the absence of any concrete evidence.

7.    In figure 2, please include information about p value and q value Bonferroni or similar parameter for multiple comparisons.

8.    In summary and conclusion, is there an evidence that population A provides lipids to population C as claimed by the authors? The substantial claims made in this section demand a strong experimental evidence.

Author Response

  1. Please re-size table 1.

Thank you for considering our paper for publication into Cells. We agree and we therefore provided a new table in order to have the right format inside of our paper.

  1. In line 152, please change “insulin growth factor” to “Insulin-like growth factor”.

    We changed the word as suggested, thank you.

  1. Please include t-SNE or UMAP plots showing subpopulations in control and lipedemic patients in figure 1.

We agree with the author that the t-SNE and UMAP plot would complete our article. However, since the manuscript is written as a communication, our objective is to introduce the topic and propose it to the scientific community interested in lipedema and reserve the data and related plots for the original article we are preparing on this topic.

  1. In figure 1, the graph appears misleading in the sense that it is conveying information that adipocyte C makes up 100% of the population in lipedema tissue, even though that is not the case. Please re-phrase the related description and re-make the graph.

Thank you for considering this part of the paper. Figure 1 was not congruent with the text. Hence, we modified the section of the manuscript with the following sentence: „Adipocyte C was exclusively found in the lipedemic fat, and not in the healthy control. The topmost highly expressed genes, the Fatty acid binding protein 4 (FABP4), CD36, Retinoid binding protein (RBP7), Serum deprivation–response protein (SDPR), alpha 2 macroglobulin (A2M) and Interferon alpha–inducible protein 27 (IFNI27), appear to be made of large [35], hypertrophic cells that are still maintaining a level of metabolic activity [36–38]. Since Adipocytes C were exclusively found in the lipedema samples, it is likely they make up the bulk of the disease tissue.

  1. Was there a difference between males and females regarding these populations? Were both sexes included in the study?

Although in our inclusion criteria we had planned to consider both female and male patients, liposuctions were performed only on female patients who have always been the largest percentage of people who present themselves in our clinic. It would be interesting to carry out the same evaluations on a sample of both male and female patients and evaluate any differences. We thank the reviewer for the question, we therefore modified the methods section explaining this detail in the manuscript.

  1. In summary and conclusion, the claim that adipocyte B is disease enabling population is not backed by results, since there is equal proportion of this population in both controls and patient subjects. It is difficult to be convinced that the population can behave in different ways in different conditions in the absence of any concrete evidence.

    Thank you for the clarification. We agree with the reviewer that the adipocyte B cannot be primarily responsible, enabling and characteristic of the disease given that we do not have sufficient results in this regard. However, in our manuscript we wrote that the adipocyte B is active in the mechanism of lymphatic destabilization, listed its possible roles in the disease considering the literature on it and concluded by saying that it could be a catalyst of the disease, a conservative prediction of the role of the adipocyte in disease. Given this considerations, we nevertheless wanted to conclude, for completeness, the section concerning the adipocyte B with the above–mentioned consideration indicated by the reviewer.

Altogether, despite this promising evaluations and the equal proportion in both controls and lipedema patients, we cannot state and consider Adipocyte B as primarily responsible of the lipedema. At present, we do not have enough data and results to make a solid evaluation of its role of Lipedema. The nature of the adipocyte has some characteristics that could catalyze or induce the disease, but the data we have collected are not sufficient to make solid or certain considerations.

  1. In figure 2, please include information about p value and q value Bonferroni or similar parameter for multiple comparisons.

Dear reviewer, we used Enrichment Scores (high and mild enrichment) to evaluate significance (see KEGG Pathway – Figure 2). In general, higher scores indicate greater overrepresentation. Hence, high enrichment correspond to a low p-value  (high p<0.01 and mild 0.010.05), indicating statistical significance. Similarly quantified as high enrichment, but with a lower enrichment score or a higher p–value compared to pathways with high enrichment. This indicates a less robust but still potentially significant association. By following these considerations, we systematically evaluated the significance of the reported data through a KEGG pathway with varying enrichment values, and prioritize pathways even for our further research.

  1. In summary and conclusion, is there an evidence that population A provides lipids to population C as claimed by the authors? The substantial claims made in this section demand a strong experimental evidence.

Dear reviewer, as expected from the typology of our article, our Communication intends to open a new sector of research, until now unexplored, on lipedema. We are aware that we do not have enough data to state with certainty that “… population A provides lipids to population C”. This is the reason why in every part of the manuscript we approached the topic with caution and without the presumption of wanting to give a certain and absolute result. So far, no responsible signaling pathways for the pathogenesis of lipedema are known, and little is generally known about the cell populations of lipedema fat. Our preliminary analysis of single-cell RNA sequencing reveals three primitive and distinct adipocyte cell populations that could play a role in lipedema. These populations could have unique gene signatures that clearly indicate a lipid-producing adipocyte, a disease-catalyst adipocyte, and a lipedemic adipocyte. Through this communication we therefore want to promote our research work (which will not stop here) in this direction and also motivate other groups to move in this area.

Round 2

Reviewer 1 Report

Comments and Suggestions for Authors

The revised manuscript has addressed many of the previous concerns and demonstrates improved results. However, a critical issue remains unresolved regarding the definition of adipocytes. Since adipocytes are buoyant after centrifugation, the analysis of the pellet (stromal vascular fraction, SVFs) from lipedema samples likely excludes most adipocyte cells. This raises significant concerns about the identification of the adipocyte cell population in the single-cell analysis. 

Clarification on how the adipocyte cell population is defined and identified in the single-cell analysis is essential for the manuscript to be considered for publication. Accurate identification and characterization of adipocytes are crucial for validating the findings and ensuring the reliability of the study's conclusions.

Author Response

Thank you for your comment which allowed us to improve the methods section of our Communication.

We acknowledge that adipocytes are buoyant and traditionally separate from the SVF during centrifugation. As in other research protocols, to ensure the inclusion of adipocytes in our single-cell analysis, we employed a modified isolation procedure that integrates both the buoyant and pellet fractions. Specifically, we collected the floating adipocytes immediately after the initial centrifugation, followed by a secondary step where the buoyant fraction was subjected to gentle enzymatic digestion to further dissociate any remaining cell clusters. This approach allowed us to obtain a comprehensive representation of both mature adipocytes and smaller pre-adipocytes or adipocyte progenitors that may sediment with the SVF. We implemented our methods section “Adipose Tissue homogenization, enzymatic cell dissociation and Single-cell RNA–sequencing” with the following statement:

To ensure the inclusion of the adipocytes which were buoyant and traditionally separate during centrifugation in our single–cell analysis, we integrated both the buoyant and pellet fractions by collecting the floating adipocytes immediately after the initial centrifugation, following by the above–mentioned secondary step where the buoyant fraction was digested to further dissociate any remaining cell clusters”.

Our method allowed us to minimize potential contamination from non-adipocyte cells. Stringent cell sorting criteria were applied during the analysis to exclude cells lacking adipocyte-specific markers. We are confident in the accurate identification of adipocyte populations in our single-cell analysis. We believe these methodological clarifications address the reviewer's concerns and demonstrate the validity of our communication.

Reviewer 2 Report

Comments and Suggestions for Authors

English language fine. No issues detected. And It can be published.

Author Response

Thank you very much for your revisions.

Best,

Andrea Pagani

Reviewer 3 Report

Comments and Suggestions for Authors

The authors have satisfactorily addressed the comments and concerns referred to in the previous review report. They have also made necessary changes to the manuscript in methods and results section that make it more complete. The reviewer agrees with the opinions of the authors regarding including more data and supporting evidence. However, the reviewer does expect the authors to perform robust experiments to support the claims made regarding to different adipocyte populations by following stringent protocols for data analysis.

Author Response

We thank the reviewer for the previous revisions who helped to significanlty improve the manuscript.